# Toxicity of Terahertz-Based Functional Mineral Water (Plant-Derived) to Immature Stages of Mosquito Vectors

**DOI:** 10.3390/insects12030211

**Published:** 2021-03-02

**Authors:** Tai-Chih Kuo, Chien-Chung Lin, Ching-Chu Tsai, Shiang-Jiuun Chen, Tso-Min Hung, Che-Chu Hsieh, Ja-Yu Lu, Rong-Nan Huang

**Affiliations:** 1Department of Biochemistry, Taipei Medical University, 250 Wu-Hsing Street, Taipei 110, Taiwan; tckuo@tmu.edu.tw; 2Department of Orthopedic Surgery, Taipei City Hospital, Taipei 100, Taiwan; ericdoctor@gmail.com; 3Department of Special Education, University of Taipei, Taipei 10048, Taiwan; 4Department of Life Science, College of Life Science, National Taiwan University, Taipei 106, Taiwan; Assumptus@gmail.com (C.-C.T.); hjchen@ntu.edu.tw (S.-J.C.); 5Department of Entomology, College of Bioresources and Agriculture, National Taiwan University, Taipei 10617, Taiwan; zewi633@gmail.com; 6Department of Photonics, National Cheng Kung University, Tainan 70101, Taiwan; modctv10@gmail.com (C.-C.H.); jayu@mail.ncku.edu.tw (J.-Y.L.); 7Master Program for Plant Medicine, College of Bioresources and Agriculture, National Taiwan University, Taipei 10617, Taiwan

**Keywords:** terahertz radiation, *Aedes aegypti*, *Aedes albopictus*, *Culex quinquefasciatus*, nonchemical pesticides, bioinsecticide

## Abstract

**Simple Summary:**

In light of the shortcomings of using insecticides, there is an urgent need to explore alternative compounds that are effective for mosquito control with minimal adverse effects. The terahertz-based functional mineral water used in the current study exhibited concentration-dependent toxicity to mosquito larvae, pupae and larvivorous predatory copepods and could be a potential biodegradable and eco-friendly bioinsecticide.

**Abstract:**

Functional mineral water and related products are popular in some Asian countries as health drinks and, recently, have been employed in agricultural crop production as well as pest control. This study aimed to investigate the survival of mosquito vectors exposed to plant-derived functional mineral water produced by terahertz technology. The terahertz-based functional mineral water used in the current study not only decreased the hatching of *Culex quinquefasciatus* (Say) larvae but also showed concentration-dependent toxicity to the 3rd instar larvae and pupae of the three mosquito species tested. *Aedes albopictus* (Skuse) and *Cx. quinquefasciatus* pupae were more susceptible to terahertz-based functional mineral water than the larval stage, as indicated by their lower LC_50_. Lower concentrations (<100 ppm) of terahertz-based functional mineral water were not lethal to the pupae; however, these low concentrations still resulted in a reduced adult emergence. Although terahertz-based functional mineral water did not significantly affect *Aedes aegypti* (Linnaeus) hatching, it could potentially be used for vector control at the larvae and pupae stages. The larvicidal and pupicidal activity of diluted terahertz-based functional mineral water gradually diminished after 24 h, indicating that it is a biodegradable and eco-friendly bioinsecticide. However, as the terahertz-based functional mineral water is also toxic to larvivorous predatory-copepods, it should not be utilized in aquatic environments where predatory-based mosquito control programs are employed.

## 1. Introduction

Alternative management strategies for mosquito vectors are urgently needed because of their ability to adapt and thrive in changing conditions. Mosquito bites induce not only physical and psychological discomfort to the affected individuals but also transmit pathogens that cause diseases, such as malaria, yellow fever, Japanese encephalitis, and dengue fever, to humans [1,2]. It is estimated that more than a half-million people die of mosquito-borne diseases annually, and the social costs are enormous [3,4]. The overuse of synthetic insecticides to eliminate mosquitoes is not considered a wise approach because it drives the emergence of pesticide-resistant mosquitoes, kills beneficial insects, and pollutes the environment. In light of the shortcomings of using synthetic insecticides, it is urgent to explore alternative compounds effective for mosquito control with minimal adverse health and environmental effects. Naturally occurring agents such as predators [5], herbal extracts [6] or microbial mosquitocides [7] have become increasingly attractive and profitable alternatives as they are recognized as green pest-control agents, possibly with novel modes of action. For example, guppies (*Poecilia reticulata*) are frequently introduced to both natural and artificial water bodies as biological control agents for mosquitoes [8,9]. Another example is functional water produced from plant-derived minerals, or acid electrolyzed water, which has proven useful in agricultural pest control and medicinal disinfection [10,11,12,13].

In this study, we investigated the effects of different concentrations of terahertz-based functional mineral water (plant-derived) on three species of common vector mosquitoes (*Aedes aegypti* (Linnaeus)*, Aedes albopictus* (Skuse)*,* and *Culex quinquefasciatus* (Say)). Functional water is usually tap water with added ingredients, such as herbs, vitamins, antioxidants, and minerals, which are said to bring health benefits. Functional water could also be physically modified mineral-containing water. For example, micro-bubbled mineral-containing water has been widely used in the decomposition of organic chemicals, wastewater treatment, medical technology and food processing [12]. Electrolytically generated acidic functional water, obtained by electrolyzing low saline concentrations, is widely used in clinical disinfection [14].

Terahertz radiation is electromagnetic radiation increasingly being applied in areas such as homeland security and medical imaging [15,16]. Recently, terahertz technology has also been applied in agricultural pest management, crop germination, and crop strength enhancement [11,17]. For crop protection, terahertz-based functional mineral water must be applied directly to the agricultural pests to cause mortality [13]. From this perspective, it would be more efficacious for mosquito-vector management since all the immature mosquito stages occur in the aquatic environment. Thus, we explored the ovicidal, larvicidal, and pupicidal effects of the terahertz-based functional mineral water against the aforementioned three mosquito species. Moreover, the functional mineral water used in this study was produced by terahertz technology, whereby plant-derived minerals (such as calcium and magnesium) are transformed into electrically charged mesoscopic crystals. These crystals can exhibit terahertz properties with wavelengths between 1 and 10 nm, splitting water molecules into hydrogen ions and hydroxide [18]. Whether the radiation properties of terahertz-based functional mineral water related to its mosquitocidal activity also was addressed in these studies.

## 2. Materials and Methods

### 2.1. Terahertz-Based Functional Mineral Water

The functional mineral water used in this study was produced by terahertz technology by Santa Mineral Co. Ltd., Tokyo, Japan [13]. The components of the functional water were plant-derived minerals and its minor and major elements as follows: silicon (0.03%), iron (0.02%), manganese (0.005%), boron (0.003%), zinc (0.002%), vanadium (0.0015%), selenium (0.001%), copper (0.0004%), calcium (1%), potassium (0.7%), magnesium (0.5%) and sulfur (0.5%). The concentrations of terahertz-based functional mineral water selected for testing in this study were based on a preliminary screening toward various stages of different mosquito species.

### 2.2. Mosquito Rearing

Laboratory colonies of *Aedes*, which had no exposure to insecticides for over 10 generations and originally were collected from Taipei (*Ae. albopictus*) and Kaohsiung (*Ae. aegypti*) of Taiwan, were reared in a growth chamber maintained at 26.0 ± 2.0 °C and 70 ± 5% relative humidity. The light regime was L 12:D 12. Batches of *Aedes* eggs were placed in plastic cups containing reverse osmosis (RO) water under vacuum aspiration overnight. The hatched larvae were reared under standard conditions as described and fed with aquarium fish food to pupation. Afterward, pupae were collected into new plastic cups with RO water and placed in cloth cages (30 × 30 × 30 cm flight cages) where they emerged. Emerging adult mosquitoes were kept in the same cages, provided with a sugar solution (10% sucrose) and RO water. To obtain the next generation of eggs, 7–10-day-old female mosquitoes were fed pig blood, according to Luo [19]. *Culex quinquefasciatus* was reared in the same conditions and fed according to Gerberg [20].

### 2.3. Laboratory Bioassay

The effects of terahertz-based functional mineral water on mosquito pupae or larvae were evaluated with a slight modification of the World Health Organization (WHO) protocol [21] at room temperature (26–28 °C). Briefly, the terahertz-based functional mineral water was diluted with RO water to achieve final concentrations of 38–333 ppm. Then, 30 larvae (third instar) or pupae (1 day after pupation) of *Ae. aegypti, Ae. albopictus*, and *Cx. quinquefasciatus* were introduced into each concentration of terahertz-based functional mineral water (100 mL) in a 250 mL beaker. Six replicates were performed for each treatment, and the experiments were repeated at least five times independently. Reverse osmosis water was used for the control group of each treatment. Mortality of the larvae and pupae was recorded over 24 h. For time-course experiments, the cumulative mortality of larvae was recorded every hour in the first 3 h, followed by 3, 6 and 12 h intervals. To test the residual activity, 250 ppm of terahertz-based functional mineral water was prepared 0–3 days ahead of bioassay. Dead larvae or pupae could be readily distinguished by their failure to swim to the surface to obtain air or lack of response to a stimulus with a fine brush (Pentel ZBS1-0).

### 2.4. Effects of Terahertz-Based Functional Mineral Water on Hatch Rate of Mosquito Eggs

Fifty freshly laid eggs of *Ae. aegypti* and *Ae. albopictus* were placed in the wells of a 24-well plate and incubated at 0–100,000 ppm of terahertz-based functional mineral water, with six replicates for each concentration. Counting of eggs was done under a stereomicroscope to evaluate viability [22,23]. The plates with *Aedes* eggs were incubated at 25 °C for 12 h. Subsequently, the 24-well plates with *Aedes* eggs were placed under vacuum aspiration for 4 h and hatched, and alive larvae were examined under a microscope.

For *Cx. quinquefasciatus*, newly laid egg rafts were placed in a 12-well plate, and the number of eggs per raft was counted under a microscope. The 12-well plate with egg rafts was incubated at 25 °C with a 12 L:12 D photoperiod. Hatched and alive larvae were recorded after 18 h as described above.

### 2.5. Effect of Functional Mineral Water on Mesocyclops Aspericornis

The monoculture of larvivorous copepods *M. aspericornis* (a freshwater copepod species) used in the experiment was obtained from the Institute of Environmental Health, College of Public Health, National Taiwan University. The experiment was carried out in a 6-well plate in which 15 adults of *M. aspericornis* were introduced into each well containing 0–330 ppm of diluted functional mineral water. The plates were then incubated at 27 ± 1.5 °C in 12 L:12 D photoperiodic conditions, and the mortality of *M. aspericornis* was recorded after 24 h. Three independent experiments were performed with six replicates each.

### 2.6. Optical Properties of Terahertz-Based Functional Mineral Water

Since the functional mineral water used in this study was produced by terahertz technology, its optical transmittance was measured with a terahertz (THz) spectroscopic system. The THz radiation in the system was synchronously pumped and probed by a mode-locked pulse laser on one pair of photoconductive antennas based on the principle of THz time-domain spectroscopy (THz-TDS). The THz radiation spectrum used to measure one sample was in the frequency range of 0.1–1 THz, with a power signal-to-noise ratio of 10^5^ and a center frequency of 0.3 THz [24]. To measure a liquid sample of terahertz-based functional mineral water with the 3 mm-wide spot size of 0.1–1.0 THz radiation, one microfluidic chamber with a 0.13 mm thickness, a 30 mm length, and a 15 mm width was prepared on a polyethylene (PE) substrate [25]. After mechanical sealing by another 2 mm-thick PE slab [25], the sample-loaded microfluidic chamber was located at the focused beam spot of radiation and measured for the transmitted power, denoted as *P_sample + cell_*. The liquid sample was then drawn out of the microfluidic chamber without any residue to obtain the transmitted radiation power, denoted as *P_cell_*. The 0.1–1.0 THz transmittance of terahertz-based functional mineral water is thus derived from the power ratio and denoted as *Tr*, *Tr* = (*P_sample + cell_*)/*P_cell_*). The spectral signal of PE substrates can be deleted, and the power signal of terahertz-based functional mineral water can be optimized from the dimensions of the microfluidic chamber [25,26]. Because the spectral transmittance of THz-TDS is performed from one shot of THz electromagnetic pulse (i.e., a wide bandwidth from 0.1 to 1.0 THz) [26], THz frequency-dependent information is reliable, and the corresponding uncertainty can be minimized in the measurement. The transmission of THz wave was calculated with *Tr* = (*P_sample_*
_+ *cell*_)/*P_cell_*), where *P_sample_*
_+ *cell*_ is the transmission of microcell loaded with terahertz-based functional mineral water, and *P_cell_* indicates the transmission of a void microcell [18].

### 2.7. Statistical Analysis

The mortality data were subjected to a probit analysis using the R package *ecotox* [27] to calculate the lethal concentration (LC_50_) required to kill 50% of the larvae or pupae with the corresponding 95% confidence intervals used according to Finney [28]. When the control mortality was between 5% and 20%, the values in the treated groups were adjusted using Abbott’s formula. The related data were discarded and repeated if control mortality was greater than 20% [29]. For each species, the mortality of larvae and pupae and hatch percentage of larvae were tested by one-way analysis of variance (ANOVA). Normality was checked by inspecting residual plots. No apparent trend, but a trace of heteroscedasticity was found in some models. We double-checked the significance of each one-way analysis by performing the Kruskal–Wallis rank-sum test, and the p-values of all models are smaller than 0.01. Therefore, the results of one-way ANOVA were reported. The differences among the concentration levels were further assessed by Tukey’s honestly significant difference (HSD) multiple comparison test. A probability level of *P* < 0.05 was used to test for significant differences between values that were expressed as means ± SE. All statistical analyses were performed in R version 3.6.1 [30]

## 3. Results

### 3.1. Larvicidal Activities of Terahertz-Based Functional Mineral Water

Figure 1 shows the 24 h larvicidal activity of terahertz-based functional mineral water. The larvae of all three mosquito species were susceptible to terahertz-based functional mineral water in a concentration-dependent manner. In particular, the larvae of *Ae. aegypti* showed 5.93% mortality at 125 ppm of terahertz-based functional mineral water and reached 98.89% mortality at 200 ppm (with LC_50_:149.8 ppm). The larvae of *Cx. quinquefasciatus* were more tolerant of the terahertz-based functional mineral water since they did not show significant mortality until 200 ppm (with LC_50_:227.9 ppm) (Table 1). The larval mortality of both *Ae. albopictus* and *Ae. aegypti* reached 95.55% and 100%, respectively, at 333 ppm of terahertz-based functional mineral water. Mortality was only 80.7% in *Cx. quinquefasciatus* under the same conditions. Although not statistically different, *Cx. quinquefasciatus* was still more tolerant than *Aedes* to the terahertz-based functional mineral water.

Figure 2 shows mortality over 24 h of mosquito larvae at various concentrations of terahertz-based functional mineral water. High concentrations of terahertz-based functional mineral water could effectively kill mosquito larvae in a short period. For example, at 333 ppm, terahertz-based functional mineral water resulted in 100% mortality of *Ae. aegypti* larvae within one hour, and *Ae. albopictus* and *Cx. quinquefasciatus* larvae reached 80% mortality within two hours. The larval mortality increased significantly in a time course-dependent manner in new prepared terahertz-based functional mineral water (Figure 2); however, the larvicidal activity of diluted terahertz-based functional mineral water diminished after 24 h (Figure 3).

### 3.2. Pupicidal Activities of Terahertz-Based Functional Mineral Water

The 24 h mortality rate of mosquito pupae in terahertz-based functional mineral water is shown in Figure 4. Although in a non-feeding stage, the mosquito pupae exhibited similar or even higher sensitivity to terahertz-based functional mineral water. The pupae of *Ae. aegypti* and *Cx. quinquefasciatus* showed 17.40% and 21.85% mortality rates at 111 ppm and reached 83.33% and 98.15% mortality rates at 250 ppm of terahertz-based functional mineral water, respectively (with LC_50_ at 130.6 ppm and 142.4 ppm, respectively) (Figure 4A and Table 2). The pupae of *Ae. albopictus* were more susceptible to terahertz-based functional mineral water, as they exhibited mortality at a lower concentration of 38 ppm (Figure 4B). Pupal mortality of *Ae. albopictus* reached 81.48% at 167 ppm, with LC_50_ at 89.6 ppm (Table 2).

Figure 5 shows the time course of pupal mortality at various concentrations of terahertz-based functional mineral water. At the highest tested concentration (250 ppm), mosquito pupae exhibited significant mortality within 6 h and reached 80% or higher mortality by 24 h. The concentration range used for pupae treatment was lower than that for larvae, particularly for *Ae. albopictus*, indicating that the pupal stage of the mosquitoes was the most susceptible to terahertz-based functional mineral water.

### 3.3. Effect of Terahertz-Based Functional Mineral Water on Mosquito Eggs

The effect of terahertz-based functional mineral water on the hatching rate of mosquito larvae is displayed in Figure 6. The eggs were the least susceptible to terahertz-based functional mineral water. Although, terahertz-based functional mineral water did not affect the hatching rate of *Ae. aegypti* larvae, it still inhibited the hatching rate of *Cx. quinquefasciatus* larvae in the presence of 100−100,000 ppm in a concentration-dependent manner. The hatching rate of *Ae. albopictus* larvae only significantly decreased when treated with higher concentrations of terahertz-based functional mineral water (10,000 and 100,000 ppm) (Figure 6A). Nonetheless, no hatched larva of any species could survive at concentrations higher than 100 ppm (Figure 6B).

### 3.4. Toxicity of Terahertz-Based Functional Mineral Water to Mesocyclops Aspericornis

The toxic specificity of terahertz-based functional mineral water was examined using adult *M. aspericornis*, a mosquito larvae predator. The results showed that the mortality rates of *M. aspericornis* were 88.5% and 98.8% when reared in an environment containing 125 ppm and 333 ppm of terahertz-based functional mineral water for 24 h, respectively.

### 3.5. Optical Properties of Terahertz-Based Functional Mineral Water

Figure 7 shows the optical properties of terahertz-based functional mineral water at different dilutions in the frequency range of 0.1–1.0 THz. Liquid water has long been considered a strong absorbent for terahertz radiation [31]. As shown in Figure 7, the undiluted terahertz-based functional mineral water exhibited a higher transmission rate of terahertz waves than pure RO water; however, the transmission rate significantly decreased when diluted with RO water. The diluted terahertz-based functional mineral water (125 ppm) showed a lower transmission of terahertz waves only in the range of 0.7–1.0 THz; however, it exhibited a lower transmission rate than water in the frequency of 0.1–1.0 THz at 167–500 ppm. In particular, the transmission rates of terahertz-based functional mineral water at 167–500 ppm were lower than that of RO water at a frequency of less than 0.6 THz. The discrepancies in transmission rate between the diluted terahertz-based functional mineral water and pure RO water diminished after 24 h (data not shown). These results were consistent with the loss of larvicidal and pupicidal activity a day after the dilution of terahertz-based functional mineral water.

## 4. Discussion

Recently, functional water has been used in many agricultural and medicinal applications [10,11,12]. In the present study, our findings indicated that terahertz-based functional mineral water was also toxic to the larvae and pupae of the three tested mosquito species in a concentration-dependent manner (0–333 ppm). The mosquitocidal activity of the terahertz-based functional mineral water was higher than the lethal effect on other pests since it is usually diluted by a factor of 300 (3333 ppm) and sprayed directly on insect pests, such as scale insects, aphids, and white-backed plant-hoppers [13]. As shown in Table 1, the larvicidal LC_50_ of terahertz-based functional mineral water ranged around 150−230 ppm, which is equivalent to that of β-caryophyllene, a natural bicyclic sesquiterpene that is a constituent of many essential oils [32]. However, its toxicity was far less than the conventional synthetic insecticides, such as temephos and fenthion with LC_50_ in ppb level [33].

Although non-feeding, the pupal stage was more susceptible to terahertz-based functional mineral water than the larval stage of each mosquito species, especially for *Ae. albopictus*, in that half the individuals were dead after exposure to more than 100 ppm of terahertz-based functional mineral water (compare Figure 1 and Figure 4). The possible explanation is that the lower metabolite rate of pupae also renders it more vulnerable to external stress. The greater sensitivity of pupae than larvae is also reported in the insect growth regulator and mosquitocidal metabolite of *Pseudomonas fluorescens* [34,35,36]. Lower concentrations of terahertz-based functional mineral water (<100 ppm) were not significantly lethal to the pupae of the mosquitoes; however, the lower concentrations still affect the survival rate following emergence (around 70% versus 100% in the control group). However, the egg stage of the mosquitoes was relatively insensitive to terahertz-based functional mineral water, the hatching percentage of *Cx. quinquefasciatus* eggs significantly decreased when placed in higher concentrations of terahertz-based functional mineral water, whereas there was no significant effect on *Ae. aegypti* eggs even when exposed to 100,000 ppm of terahertz-based functional mineral water (Figure 6A). The susceptibility of mosquito eggs to terahertz-based functional mineral water was in the order *Cx. quinquefasciatus > Ae. albopictus > Ae. aegypti* (Figure 6A). This phenomenon is probably correlated with their resistance to desiccation, i.e., the eggs of *Ae. aegypti*, *Ae. albopictus*, and *Cx. quinquefasciatus* can survive outside the water for long, intermediate, and short periods, respectively, due to varying extents of melanization and serosal cuticle formation [37]. Therefore, melanization, as well as the serosal cuticle of the eggshell, may play an important role in the sensitivity of mosquito eggs to terahertz-based functional mineral water.

The present findings also showed that the larvicidal activities of terahertz-based functional mineral water diminished a day after being diluted with RO water (Figure 3). The pH of stock terahertz-based functional mineral water is above 4 and ranges from 5.88–6.12 after 3000–10,000-fold (330–100 ppm) dilution. The pH change is unlikely to be the factor causing the larvicidal/pupicidal activities of terahertz-based functional mineral water since *Ae. aegypti* can complete larval development in waters ranging from pH·4 to pH·11 [38,39]. The reduction in larvicidal activity coincided with diminishing discrepancies in transmission rate between the diluted terahertz-based functional mineral water and RO water in the frequency range of 0.1–1.0 THz. These results indicated that the terahertz treatment might play a role in the toxicity of functional mineral water against mosquito vectors since the minerals in the functional water are transformed into electrically charged mesoscopic crystals by terahertz technology. These crystals might exhibit terahertz properties with wavelengths between 1 and 10 nm, splitting water molecules into hydrogen ions and hydroxide [18], which exert toxic effects toward mosquito vectors. However, this assumption needs to be further verified.

As terahertz-based functional mineral water contains only water and common minerals, it is considered harmless to the human body and may be used as an eco-friendly material to replace traditional chemical drugs or therapy [17,40]. Functional mineral water or related products produced by terahertz technology are popular in Japan as health drinks. Thus, there is speculation as to whether terahertz-wave treated mineral water could be an alternative to chemical pesticide applications in crops for pest management as well as germination and crop strength enhancement. For general pest control, the terahertz-based functional mineral water must be ingested or sprayed where the pests are found (i.e., the terahertz-based functional mineral water is a stomach or contact poison). From this perspective, terahertz-based functional mineral water would be more effective against aquatic insects, particularly the mosquito vector, since most of its life stages (the immature stages) occur in the aquatic environment.

The present study showed that terahertz-based functional mineral water could effectively knock down mosquito vectors extremely rapidly at the highest concentration (Figure 2 and Figure 5). This fast insecticidal activity renders terahertz-based functional mineral water as a suitable alternative for conventional synthetic insecticides. The potential challenge for the application of terahertz-based functional mineral water against mosquito vectors may be the high LC_50_ beyond 100 ppm as compared with that of traditional insecticides; however, terahertz-based functional mineral water only contained common minerals that are affordable and easily degraded in the environment. Therefore, terahertz technology-based terahertz-based functional mineral water could be categorized as an eco-friendly bioinsecticide. However, our findings showed that it is also toxic to larvivorous predatory-copepods; therefore, it should not be utilized in aquatic environments simultaneously with biocontrol programs using predatory-copepods.

## Figures and Tables

**Figure 1 insects-12-00211-f001:**
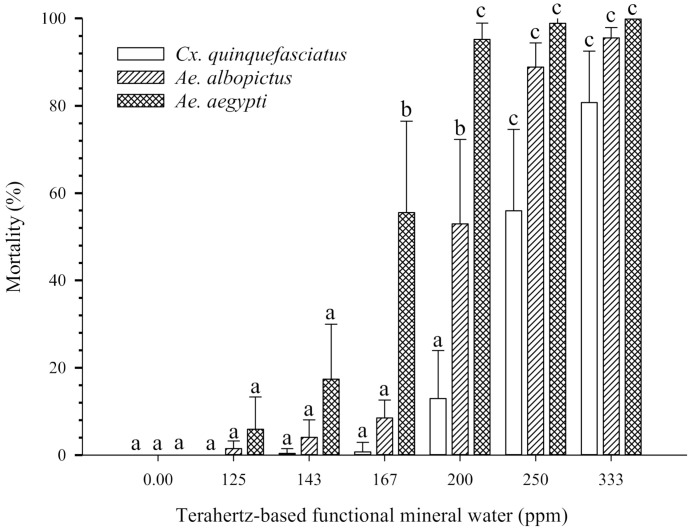
Effects of terahertz-based functional mineral water on mosquito larvae. The third instar larvae were exposed to various concentrations of terahertz-based functional mineral water, and the mortality of the larvae was recorded after 24 h. Data are presented as mean ± SE values from at least three independent experiments, and different letters indicate significant differences (*p* < 0.05) between different concentrations at each species.

**Figure 2 insects-12-00211-f002:**
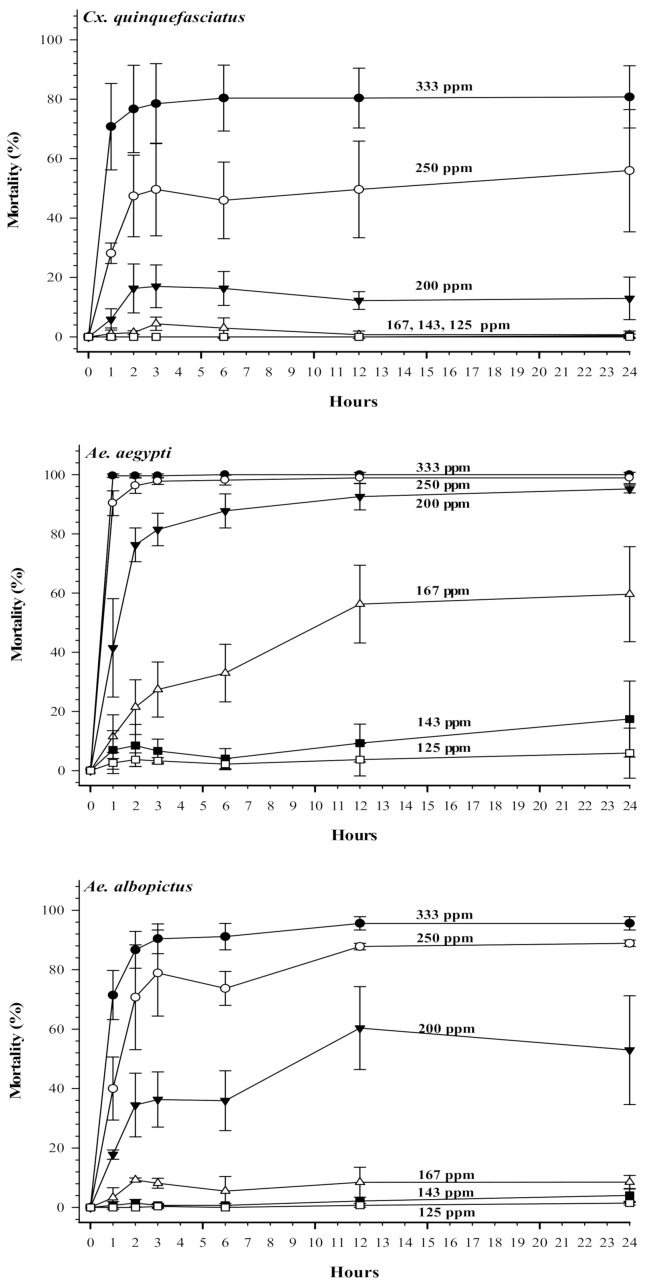
Time effects of terahertz-based functional mineral water on mosquito larval mortality (%). The third instar larvae were reared in reverse osmosis (RO) water containing various concentrations of terahertz-based functional mineral water, and mortality of larvae was recorded at time points indicated on the x-axis. Data are presented as mean ± SE values from at least three independent experiments, and different letters indicate significant differences (*p* < 0.05) between different concentrations at each species.

**Figure 3 insects-12-00211-f003:**
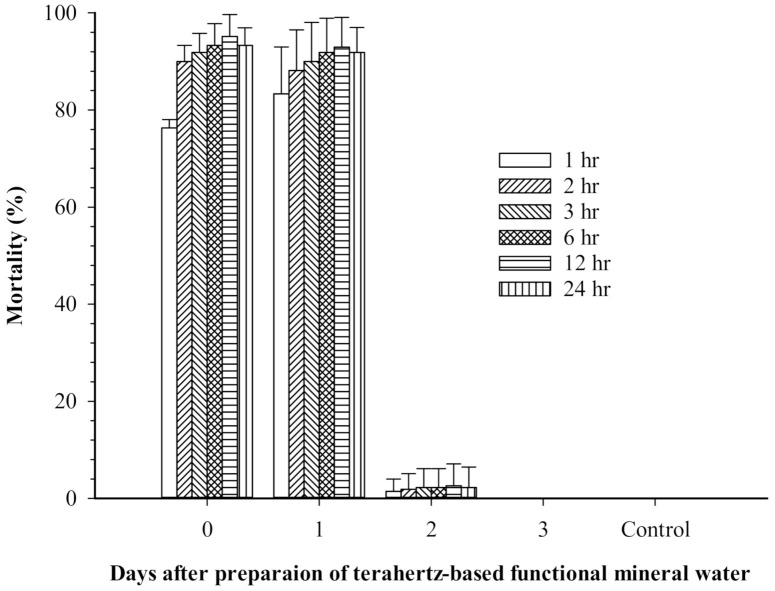
Residual activity of terahertz-based functional mineral water toward Aedes aegypti larvae. The Ae. aegypti larvae were transferred to a solution containing 250 ppm terahertz-based functional mineral water, which was prepared 3, 2, 1 days ahead and the day (0 days) of the experiment. Afterward, the mortality of larvae was recorded after 24 h. The control group was reared in RO-only water simultaneously. Data are presented as mean ± SE values from at least three independent experiments.

**Figure 4 insects-12-00211-f004:**
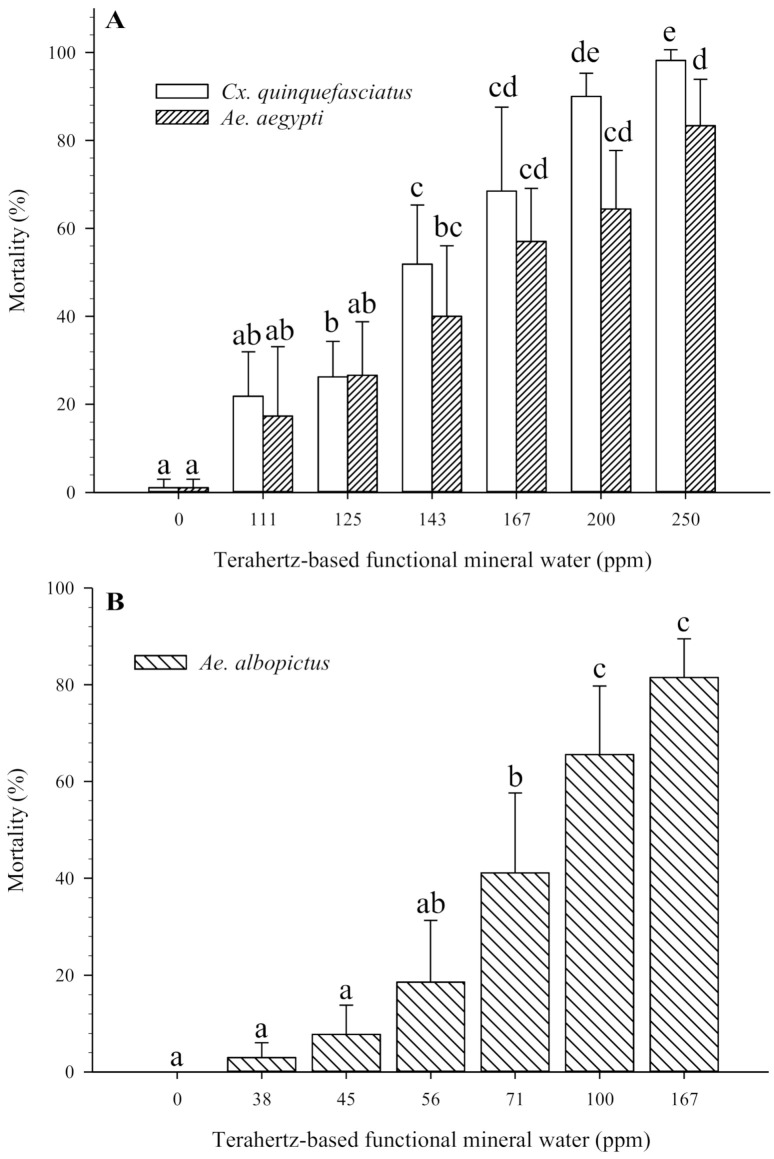
Effects of terahertz-based functional mineral water on mosquito pupae mortality (%). On the second day of pupation, the mosquito pupae were transferred to a solution containing various concentrations of terahertz-based functional mineral water, and pupal mortality rates were recorded after 24 h. Data are presented as mean ± SE values from at least three independent experiments, and different letters indicate significant differences (*p* < 0.05) between different concentrations at each species.

**Figure 5 insects-12-00211-f005:**
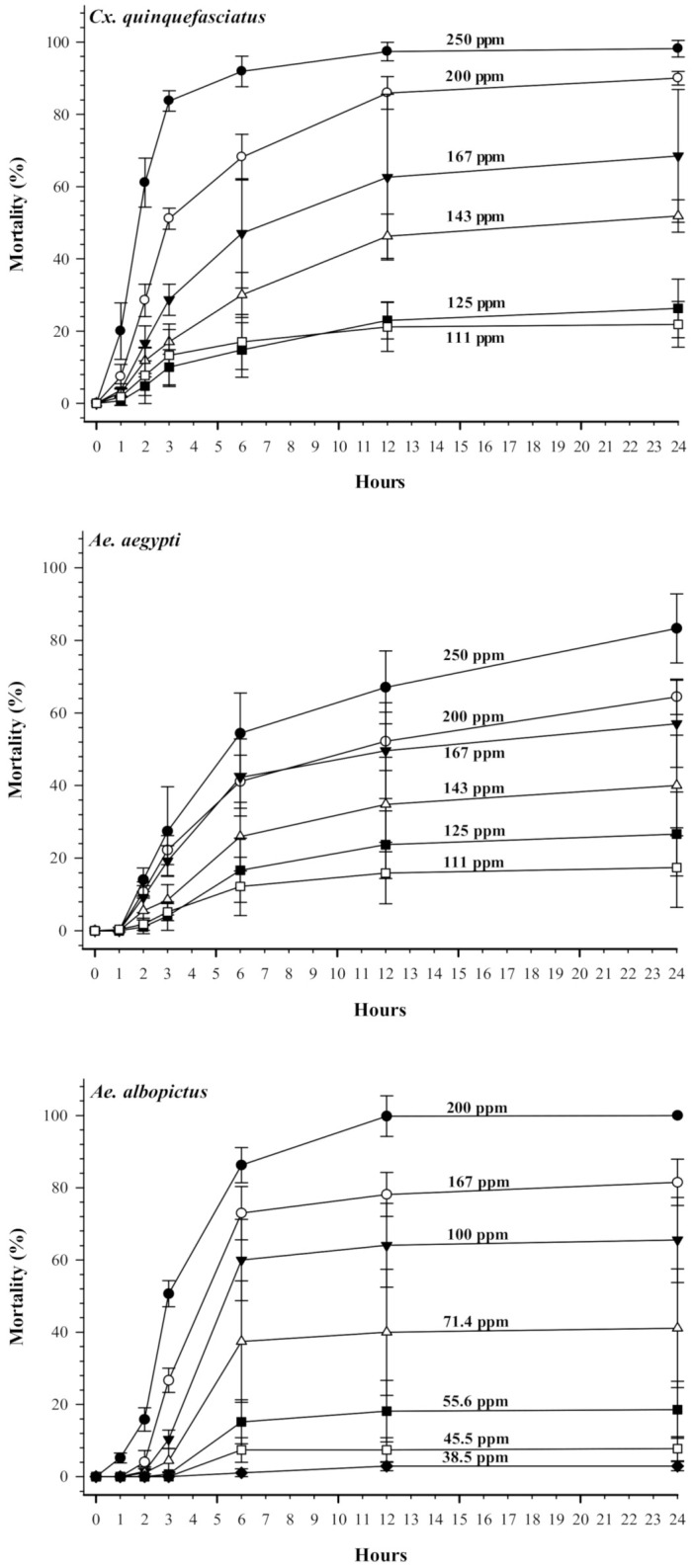
Time effects of terahertz-based functional mineral water on mosquito pupal mortality (%). On the second day of pupation, the mosquito pupae were transferred to a solution containing various concentrations of terahertz-based functional mineral water. The pupal mortality rate was recorded at various time points as indicated on the x-axis. Data are presented as mean ± SE values from at least three independent experiments, and different letters indicate significant differences (*p* < 0.05) between different concentrations at each species.

**Figure 6 insects-12-00211-f006:**
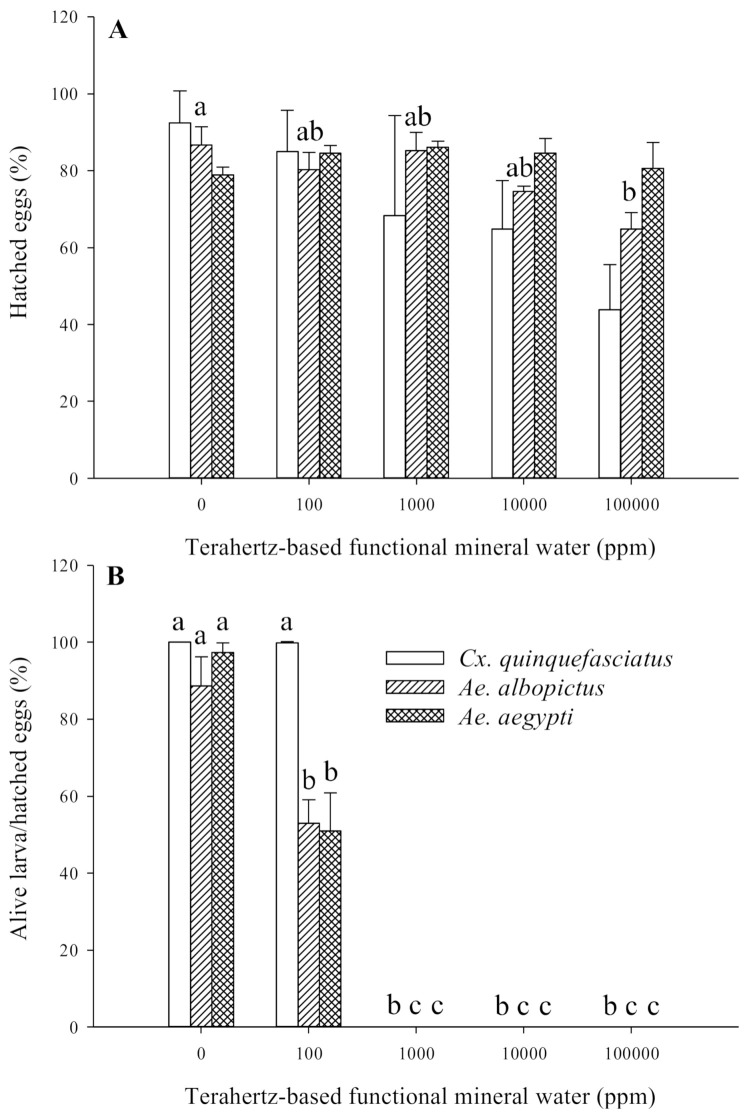
Effects of terahertz-based functional mineral water on the hatching rate of mosquito larvae. The *Ae. aegypti and Ae. albopictus* eggs were immersed in various concentrations of terahertz-based functional mineral water and aspirated. For *Culex. quinquefasciatus*, newly laid egg rafts were incubated in varying concentrations of terahertz-based functional mineral water. Afterward, hatching (**A**) and larvae survival (**B**) were recorded accordingly. Data are presented as mean ± SE values from at least three independent experiments, and different letters indicate significant differences (*p* < 0.05) between different concentrations at each species.

**Figure 7 insects-12-00211-f007:**
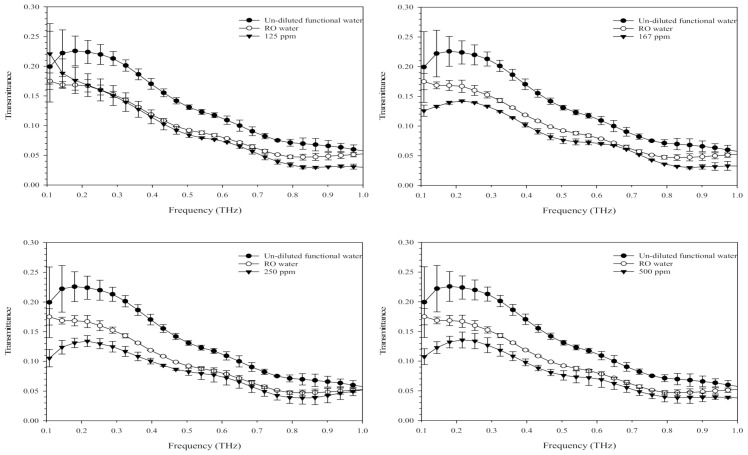
Optical properties of the terahertz-based functional mineral water. The terahertz-based functional mineral water was diluted with RO water and scanned with terahertz time-domain spectroscopy in the frequency range of 0.1–1 THz.

**Table 1 insects-12-00211-t001:** Toxicity of terahertz-based functional mineral water to larvae of vector mosquitoes.

	*Aedes aegypti*	*Aedes Albopictus*	*Culex quinquefasciatus*
LC_50_ (ppm) ^a^	149.8	183.4	227.9
(LCL-UCL) ^b^	(141.1–159.0)	(165.7–203.0)	(205.0–253.5)
LC_90_ (ppm) ^a^	198.0	268.4	345.6
(LCL-UCL) ^b^	(187.2–218.4)	(250.3–298.1)	(318.3–389.4)
Regression line	y = −28.41 + 15.11x	y = −20.29 + 10.94x	y = −18.69 + 9.83x
Slope	15.10 ± 2.33	10.93 ± 1.54	9.85 ± 1.39
Chi-squared value	1.9215	6.2926	2.9058

^a^ LC: lethal concentration. ^b^ 95% confidence interval, LCL, and UCL indicate lower and upper confidence levels.

**Table 2 insects-12-00211-t002:** Toxicity of terahertz-based functional mineral water to pupae of vector mosquitoes.

	*Aedes aegypti*	*Aedes albopictus*	*Culex quinquefasciatus*
LC_50_ (ppm) ^a^	130.6	89.6	142.4
(LCL-UCL) ^b^	(123.2–138.5)	(57.8–78.2)	(118.4–130.7)
LC_90_ (ppm) ^a^	286.2	179.2	204.4
(LCL-UCL)^b^	(250.1–354.4)	(150.2–232.1)	(193.1–219.6)
Regression line	y = −6.75 + 5.30x	y = −3.33 + 4.27x	y = −13.14 + 8.41x
Slope	5.30 ± 0.89	4.28 ± 0.57	8.41 ± 1.12
Chi-squared value	0.5717	3.0957	0.8243

^a^ LC: lethal concentration. ^b^ 95% confidence interval, LCL, and UCL indicate lower and upper confidence levels.

## Data Availability

Not applicable.

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
