# Peer review of "Toxicity of Terahertz-Based Functional Mineral Water (Plant-Derived) to Immature Stages of Mosquito Vectors"

_insects, 2021, doi:10.3390/insects12030211_

Round 1

Reviewer 1 Report

Dear authors,

Thank you for considering my comments. However, it seems that one point has not appropriately addressed in the revised manuscript. Could you please clarify if you recorded the hatched eggs and then the hatched larvae? If this is the case, you should use in the text the term “Alive larvae/hatched eggs” instead of “Alive larvae/hatched larvae”. Also, y axis in fig 5. should be amended as follows: “Hatched eggs (%)” for the first figure and “Alive larvae/hatched eggs (%)” for the second figure.

Author Response

Reviewer 1:

Thank you for considering my comments. However, it seems that one point has not appropriately addressed in the revised manuscript. Could you please clarify if you recorded the hatched eggs and then the hatched larvae? If this is the case, you should use in the text the term “Alive larvae/hatched eggs” instead of “Alive larvae/hatched larvae”. Also, y axis in fig 5. should be amended as follows: “Hatched eggs (%)” for the first figure and “Alive larvae/hatched eggs (%)” for the second figure.

Response: We have revised our manuscript accordingly (Page 5, Line136 & 140-141; Figure 6).  Thanks for your comments.

Reviewer 2 Report

The authors have made an effort to address the previous questions raised, but there is still a key problem that I will leave the editor to have a final say on. Otherwise I am satisfied with the changes made. 

Author Response

Reviewer 2:

The authors have made an effort to address the previous questions raised, but there is still a key problem that I will leave the editor to have a final say on. Otherwise I am satisfied with the changes made. 

Response: Thanks

Reviewer 3 Report

Authors address all of this reviewer's comments properly.

Author Response

Reviewer 3:

Authors address all of this reviewer's comments properly.

Response: Thanks

Reviewer 4 Report

English language needs re-revision and any missing methods need to be added.

Specific comments:

No true objectives - need to add.

Elements should not have capital letters

What water type was used, you changed from deionized to reverse osmosis. Do you mean the earlier deoionized was a mistake and it was reverse osmosis all along?

Degree symbols look weird still

Still no mention of where or how the pupae are obtained - need to extend methods

What happened to the controls? Were they used in the analysis? Add a sentence to the analysis section

What are the bars and errors in all the graphs? Means and standard errors?? Add info to each legend.

The methods of the time course experiment have not been described so it is not clear if you interpretation is justified but i dont think it is.

Non significant results need to be explained with more care

A new section in the results on mesocyclops mortality was added, all the corresponding methods need to be added, rearing, bioassay, stats.

Why was the transmittance experiment done? It is still unclear to me (adding objectives to the introduction would help).

Some ranges still with tilda rather than hyphen or endash

What is a stomach poison?

Reference suggestions were not addressed

Author Response

Reviewer 4

Comment: English language needs re-revision and any missing methods need to be added.

Response: We have followed the reviewer’s comments to revise our manuscript.  We have also asked for professional editing of our manuscript before submitting (please see the attached certificate of English editing).  We have also added necessary methods (see response to specific comments below).  

Specific comments:

Comment: No true objectives - need to add.

Response: Since terahertz radiation is increasingly being applied in various areas and the functional mineral water used in this study was produced by terahertz technology whereby plant-derived minerals (such as calcium and magnesium) are transformed into electrically charged mesoscopic crystals.  To explore how is the radiation property of terahertz-based functional mineral water related to its mosquitocidal activity, we measured the optical properties of terahertz-based functional mineral water with a terahertz (THz) spectroscopic system.  We have added a sentence to address the purpose of these studies (Page 3, Line 87-88).  We hope it would be more clear for the objective of these studies.

Comment: Elements should not have capital letters

Response: We have revised our manuscript accordingly (Page 3, Line 95-97).  Thanks for your comment.

Comment: What water type was used, you changed from deionized to reverse osmosis. Do you mean the earlier deionized was a mistake and it was reverse osmosis all along?

Response:  Yes, the earlier deionized water was a mistake and it was reverse osmosis all along.  Thanks for your comment.

Comment: Degree symbols look weird still

Response:  We have revised our manuscript accordingly (Page 4, Line 118; Page 5, Line 135 &140).  Thanks for your comment.

Comment: Still no mention of where or how the pupae are obtained - need to extend methods.

Response:  We have revised our manuscript accordingly (Page 4, Line 108-112).  Thanks for your comment.

Comment: What happened to the controls? Were they used in the analysis? Add a sentence to the analysis section.

Response: We have added a sentence to the statistical analysis section (2.5.) (Page 6, Line 181-184).  Thanks for your comment.

Comment: What are the bars and errors in all the graphs? Means and standard errors?? Add info to each legend.

Response: We have revised our manuscript accordingly (Figure 1-5 legend).  Thanks for your comment.

Comment: The methods of the time course experiment have not been described so it is not clear if you interpretation is justified but i dont think it is.

Response: We have revised our manuscript accordingly (Page 4, Line 125-126).  Thanks for your comment.

Comment: Non significant results need to be explained with more care

Response: We have revised our manuscript accordingly (Page 7, Line 206-209).  Thanks for your comment.

Comment: A new section in the results on mesocyclops mortality was added, all the corresponding methods need to be added, rearing, bioassay, stats.

Response: We have revised our manuscript accordingly (Page 5, Line 143-151).  Thanks for your suggestion.

Comment: Why was the transmittance experiment done? It is still unclear to me (adding objectives to the introduction would help).

Response: Since terahertz radiation is increasingly being applied in various areas and the functional mineral water used in this study was produced by terahertz technology whereby plant-derived minerals (such as calcium and magnesium) are transformed into electrically charged mesoscopic crystals.  To explore how is the radiation property of terahertz-based functional mineral water related to its mosquitocidal activity, we measured the optical properties of terahertz-based functional mineral water with a terahertz (THz) spectroscopic system.  We have added a sentence to address the purpose of these studies (Page 3, Line 87-88).  We hope it would be more clear for the objective of these studies.

Comment: Some ranges still with tilda rather than hyphen or endash.

Response: We have revised our manuscript accordingly (Page 3, Line 79; Page 17, Line 433-434).  Thanks for your suggestion.

Comment: What is a stomach poison?

Response: a stomach poison is a type of pesticide that is ingested by a pest and absorbed into its body, causes its death.

Comment: Reference suggestions were not addressed

Response: We have checked the reference list again (Page 19-22).  Thanks for your suggestion.

This manuscript is a resubmission of an earlier submission. The following is a list of the peer review reports and author responses from that submission.

Round 1

Reviewer 1 Report

Dear authors,

This is a well written paper dealing with efficacy evaluation of functional mineral water against eggs, larvae and pupae of mosquitoes. The findings are interesting since they show larvicidal and pupicidal properties of the tested functional miner water  against larvae and pupae of three mosquito species, namely Ae. albopictus, Ae. aegypti and Cx. quenquefasciatus.

Further down, you will find some suggestions/comments of minor importance that may benefit the manuscript.

Lines 41-44: You could add literature references for these two sentences.

Lines 48-50: I suggest removing this specific example of biological control with cocepods, as not so relevant.

Lines 71-72: We could elaborate more on the scope of this manuscript, addressing the determination of ovicidal, larvicidal and pupicidal effect of the tested mineral water against three mosquito species, namely Ae. albopictus, Ae. aegypti and Cx. quenquefasciatus.

Lines 68-69 & 75-76: Please clarify if the tested mineral water is currently used for agriculture or other purposes.

Table 1: It is better to use just one decimal for the LC50 values and confidential intervals.

Lines 104-109 & Figure 5: It seems that regarding eggs, you recorded the hatched eggs and then you counted the hatched larvae. Please clarify, and if so it should be more clearly indicated in the text and amend y axis in fig 5. as follows: “Hatched eggs (%)” for the first figure and “Alive larvae/hatched eggs (%)” for the second figure.

Lines 177 & 224-225: From the results it is evidenced that pupae were more susceptible to the miner water than larvae, although it is a non-feeding stage. Could you please give an explanation, and add it in the discussion, for this finding using literature data regarding susceptibility of pupae on other substances?

Reviewer 2 Report

Review: Toxicity of functional mineral water (plant-derived) to immature stages of mosquito vectors

Although this manuscript presents some interesting findings, I have a few key problems with the manuscript.

Some of the more minor issues, including cases of being overly colloquial are highlighted on the manuscript. The key problems are as follows:

The concept of what you are actually using as a larvicide is not clear. The definition of functional water is nebulous at best. There are many things that seem to be called functional water, and your MS does not give an idea of where to source this water. How can this experiment be replicated? You mention that this functional water is plant-derived, but then go into detail about the radiation used (curiously without clarifying Teraherz radiation is) but there is still no information about it being plant derived. Therefore, as it is not clear what you are using as a larvicide or where to get it, this makes it difficult to replicate.

I am also concerned about the efficacy of the water. It seems to require quite high dosages, and looking at the time course experiments, the Culex larvae do not reach 100% mortality even at the high dosage. The highest dosages also do not result in complete death, so this is a little concerning.

As the larvicide is water, it would be critical that the effects of the treatment on non-target organisms. It is critical that this would have to be tested.

Therefore, although there are apparent lethal effects, the fact that the experiment cannot be replicated, that the effect of this water has not been tested on non-target organisms, the way the larvicide functions or the fact that it is not clear what the larvicide actually is, I cannot recommend publication.

Reviewer 3 Report

The manuscript describes toxicity of the functional mineral water to immature stages (egg, pupa and larva) of three mosquito species (Culex quinquefasciatus, Aedes aegypti and Ae. albopictus). The bioassay results show that the functional mineral water is toxic to all three immature stages of three mosquito species tested.

This is an interesting study. However, the mode of action of the functional mineral water is not clearly demonstrated. Since it has significant toxicity to egg, pupal and larval stages of tested mosquito species, this reviewer think that authors should include other non-target organisms in the bioassay to see if it affects them as well. 

For the mosquito species used, it would be nice to include Anophles species, another very important vector of diseases. 

For Table 1, I suggest authors make a separate table for the pupal bioassay.

Lines 136-138: These results are not statistically different and should be mentioned.

Figure 3: It seems that the graph is cut off for the mortality of Cx. quinquefasciatus at 250 ppm. Clarify.

Figure 3 and 4: Authors should include the results of Ae. albopitus pupae at 200 and 250 ppm. 

Reviewer 4 Report

I agree that novel biopesticides are urgently needed for mosquito management and the active ingredient sounds interesting. However, I have a few major concerns about the manuscript.

  1. The controls should have been done with RO water instead of or as well as deionised water. While it is clear that you do have a dose response with the functional water this is a serious error in the experimental design, to not use the diluent as the control solution.
  2. The experiment on functional mineral water is poorly described, the treatments were not defined, the water type is referred to as RO and as distilled water. It is also not made clear what the relevance this experiment is to mosquito toxicology, if none I do not think it should be included.
  3. I am concerned that while you have shown that functional water is toxic to mosquitoes we do not know what aspect of the water it is that is having the effect, given that the water is made up of a mixture of minerals.
  4. Throughout the manuscript you mention that the functional water loses efficacy after (but you mean at/by) 24 hours. However, your experiment (as far as is explained in your methods) did not test this. To test this you would need to add mosquitoes to functional water that had been aged for set time frames. Your study showed that the effect plateaued at 24 hours and after that no further mortality occurred. This does not mean that your active ingredient has lost efficacy or has degraded. If I misunderstand then the design needs to be made more clear. If not then all mention of this loss of effect needs to be rephrased.
  5. The discussion needs to compare toxicology of functional water to other pesticides, as well as toxicology of functional water to other insect species.
  6. The formatting of the manuscript needs to be checked throughout, as well as grammar and clarification of many sentences. See attached document for many suggestions to improve appearance and readability of the manuscript.
  7. Please provide your control mortality somewhere, and describe any corrections conducted in the methods.

I have many minor concerns and suggestions, which are in the attached PDF.
